# Effects of Different Fasting Interventions on Cardiac Autonomic Modulation in Healthy Individuals: A Secondary Outcome Analysis of the EDIF Trial

**DOI:** 10.3390/biology12030372

**Published:** 2023-02-26

**Authors:** Paul Zimmermann, Daniel Herz, Sebastian Karl, Johannes W. Weiß, Helmut K. Lackner, Maximilian P. Erlmann, Harald Sourij, Janis Schierbauer, Sandra Haupt, Felix Aberer, Nadine B. Wachsmuth, Othmar Moser

**Affiliations:** 1Division of Exercise Physiology and Metabolism, Department of Sport Science, University of Bayreuth, 95440 Bayreuth, Germany; 2Interdisciplinary Center of Sportsmedicine Bamberg, Klinikum Bamberg, 96049 Bamberg, Germany; 3Department of Cardiology, Klinikum Bamberg, 96049 Bamberg, Germany; 4Department of Physiology, Medical University of Graz, 8036 Graz, Austria; 5Interdisciplinary Metabolic Medicine Research Group, Division of Endocrinology and Diabetology, Medical University of Graz, 8036 Graz, Austria

**Keywords:** autonomic cardiac modulation, electrocardiography, electrocardiographic time interval, intermittent fasting, fasting intervention

## Abstract

**Simple Summary:**

The impact of a fasting intervention on cardiometabolic health is a noteworthy topic in the scientific community, whereby the coherences of a fasting intervention and autonomic cardiac responses is a topic that is scarcely analyzed. Therefore, the aim of this study is to scientifically investigate the influence of different fasting interventions on cardiometabolic health, i.e., autonomic cardiac response. Twenty-seven individuals (male 16, female 11, aged 26.3 ± 3.8 years) participated and completed this study with a controlled run-in period and were included for analyses. Following the controlled run-in period, participants were randomized into three different fasting groups: (I) alternate day fasting (24/24 h of fasting/feasting; *n* = 8); (II) the 16/8 fasting cohort (16/8 h of fasting/feasting, *n* = 11) and (III) the 20/4 fasting method, including 20/4 h of fasting/feasting (*n* = 8). An analysis of baseline electrocardiogram parameters and heart rate variability parameters following different fasting interventions demonstrated the safety of these interventions without impacting on heart rate variability parameters during Schellong-1 testing, and demonstrated comparable preserved autonomic responses independently of the fasting intervention. In conclusion, different fasting protocols showed comparable cardiac autonomic responses, determined by electrocardiogram and heart rate variability measurements.

**Abstract:**

The impact of a fasting intervention on electrocardiographic (ECG) time intervals and heart rate variability (HRV) is a focus that is scarcely analyzed. The main focus of these secondary outcome data was to describe the impact of a different fasting intervention on ECG and HRV analyses. Twenty-seven healthy individuals participated in this study (11 females, aged 26.3 ± 3.8 years, BMI 24.7 ± 3.4 kg/m^2^), including a pre-intervention controlled run-in period. Participants were randomized to one of the three fasting cohorts: (I) alternate day fasting (ADF, *n* = 8), (II) 16/8 fasting (16/8 h of fasting/feasting, *n* = 11) and (III) 20/4 fasting (20/4 h of fasting/feasting, *n* = 8). An analysis of baseline ECG parameters and HRV parameters following different fasting interventions demonstrated the safety of these interventions without impacting on heart rate variability parameters during Schellong-1 testing, and revealed comparable preserved autonomic cardiac modulation (ACM) independently of the fasting intervention. In conclusion, different short-term fasting interventions demonstrated no safety ECG-based concerns and showed comparable ACM based on ECG and HRV assessments. Finally, our research topic might strengthen the scientific knowledge of intermittent fasting strategies and indicate potential clinically preventive approaches with respect to occurring metabolic disease and obesity in healthy young subjects.

## 1. Introduction

Two-thirds of men and half of women suffer from being overweight and a quarter of German adults are obese. Therefore, the research for effective weight loss methods has been ongoing for several years [1,2]. Fasting, an integral of many religious and ethnic cultures, has emerged as a potential remedy to treat this ‘overweight and obesity crisis’ [3]. Fasting may be conducted via several different approaches, originally initiated to cleanse the body and mind as a spiritual and religious cause for Christians around Easter and Muslims during Ramadan. 

The scientific community has picked up the term ‘intermittent fasting’ (IF) that mainly means to refrain from eating for a certain period of time. Typical forms of fasting include fasting for 24 h every other day, with ad labitum (ad lib) caloric intake on the food/caloric drink intake days; this is called alternate day fasting (ADF). Nevertheless, most applied forms of IF represent time-restricted feeding periods (TRF), such as fasting for 16 h and caloric intake for the remaining 8 h (16/8). An enhanced version of IF 16/8 is the 20/4 method, during which individuals fast for 20 h with only a daily 4 h window for caloric intake. These different methods of IF and TRF depend on the type of protocol. ADF can include a “fasting day”, whereby < 25% of baseline energy needs are allowed to be consumed; alternatively, some protocols may allow for no caloric intake on fasting days at all [4]. 

IF has gained much attention due to several studies evaluating and discussing the impact of IF. Therefore, IF strategies encompass a broad spectrum of effects in the management of non-obese as well as in obese subjects.

In non-obese subjects, ADF has been shown to be feasible and has improved the average daily fat oxidation accompanied by mild fat mass loss [5]. Additionally, positive effects of the ADF strategy on cholesterol metabolism and cardioprotective effects, such as an improvement in cardiovascular parameters and lowering the risk of cardiovascular disease, were reported in normal weight adults in previous studies [6,7].

Focusing on adult men with obesity, Byrne et al. revealed positive effects of intermittent energy restrictions on weight loss and fat mass reduction [8]. In this context, a recent review from Templeman et al. compared different energy-restriction fasting methods due to the management of metabolic health, in which they found that ADF without energy restrictions was less effective in reducing body mass than daily energy restrictions [9]. In accompaniment, the impact of IF on cholesterol metabolism in obesity is differential, whereby some previous studies reported a cardioprotective reduction in low density lipoprotein (LDL) cholesterol between 7% and 32% and for total cholesterol concentration from 6% to 21%, as well as unchanged high-density lipoprotein (HDL) cholesterol concentrations [6,10,11,12]. ADF and IF have shown decreased serum glucose and insulin levels in rodents, whereby the feasibility and translation in daily routines for clinical populations is limited. Nevertheless, positive effects on reductions in fasting insulin levels ranged from 11% to 57% and improvements in insulin resistance have been revealed in overweight subjects, contributing to cardiovascular risk reduction, whereby overnight fasting blood glucose levels remained largely unchanged [13,14].

The latest research evaluating IF in people with type 2 diabetes on insulin therapy revealed significantly reduced HbA1c, body weight and insulin dose in comparison to the standard of care [15]. Additionally, fasting in general is associated with increasing levels of free fatty acids and ketone bodies, which may have opposing effects on inflammation [16]. 

Next to the presented positive effects of IF strategies and calorie restrictions on cardiovascular health by delaying the progress of cardiovascular ageing and arteriosclerosis [17,18], previous clinical research revealed, in detail, the coherences between HRV and ACM parameters’ improvement and weight lost or IF strategies, both in adults with central obesity as well as in moderately overweight postmenopausal women [17,19].

Due to the variety of studies scientifically processing the positive consequences of fasting, we initiated our study protocol, of which investigated “The effects of Different Fasting Interventions on Anthropometry, Metabolic Health and Functional Performance in non-obese Individuals: a randomized trial with a controlled run-in period–the EDIF trial”, and presented in this manuscript the secondary outcome analysis, focusing on autonomic cardiac modulation (ACM) based on electrocardiogram (ECG) and heart rate variability (HRV) assessments. The novelty of our secondary outcome EDIF analysis was to investigate how ADF (0/100% daily caloric intake), 16/8 IF (0/100% daily caloric intake) and 20/4 IF (0/100% daily caloric intake) might modify ECG baseline parameters and HRV parameters during an interventional period follow up (8 weeks) after an initial controlled run-in period of 4 weeks. We hypothesized that there would be significant differences in-between the participating individuals for the ECG baseline parameters and HRV measurements, i.e., autonomic sympathicovagal balance, due to the transiently acquired metabolic alterations during the different fasting schedules [4].

## 2. Materials and Methods

This was a secondary outcome analysis of a single-center randomized trial with a controlled run-in period. In the participating healthy, physically active individuals, we assessed the effects of ADF (0/100% daily caloric intake), 16/8 IF (0/100% daily caloric intake) and 20/4 IF (0/100% daily caloric intake) on ECG baseline parameters and HRV parameters. The study protocol was registered at the local ethics committee of the University of Bayreuth (Bayreuth, Germany) (Az. O 1305/1–GB 20 May 2022).

The trial was planned and carried out in accordance with the principles of Good Clinical Practice and the Declaration of Helsinki [20]. Potential participants were informed about the study protocol in the accompanying study center, and the participating subjects had to sign a written consent form to take part in the study before any trail related examinations were performed. This trial was conceived as a proof-of-concept study and was subscribed at the German Clinical Trials Register (DRKS00029003).

### 2.1. Inclusion and Exclusion Criteria 

The participating subjects had to meet the following inclusion criteria: male or female gender, body mass index (BMI) between 20.0 and 29.9 kg/m^2^ and aged between 18 and 65 years. After informed consent was obtained, a body mass specific oxygen uptake > 20 mL/min/kg^−1^ and normal fasting glucose tolerance were further preconditions to be enrolled in the trial. Individuals were excluded if they fulfilled one of the subsequent requirements: enrollment in another trial, receiving investigational medicinal devices, significant bradycardia tendency < 35 beats per minute (bpm) at screening or significant abnormal ECG alterations at screening—as judged by the investigator—and supine blood pressure outside of the range of 90–150 mmHg for systolic and 50–95 mmHg for diastolic after resting for five minutes in a supine position. Furthermore, participating individuals had to be withdrawn if they were suffering from multiple and/or severe allergies to drugs or foods or a history of severe anaphylactic reactions. A further exclusion criterion was the anamnesis of a life-threatening disease or clinically severe disease that could directly affect the trial results, as judged by the investigator. The common and uncommon usage of any pharmaceutical drugs, which might influence the analyzed ECG parameters, represented the exclusion criteria (i.e., antihypertensive drugs, antiarrhythmic medication, antidepressant medication with potential impact on ECG time intervals, as QT lengthening). Before being enrolled in the study, inclusion and exclusion criteria were evaluated and our participating subjects were examined by a medical investigator at the screening appointment. 

### 2.2. Trial Schedule 

Our participating subjects (*n* = 27) were randomized to the order of fasting interventions by a research associate that was not further involved in the study (performed by Research Randomizer 4.0 (Social Psychology Network, Lancaster, PA, USA)® (1:1:1) [21]. This randomized study was designed with a controlled initial run-in period of 4 weeks, starting with the screening visit and being followed by two study-specific visits, as described below (Figure 1). The controlled run-in period of 4 weeks was evaluated at visit 2. Afterwards, the ECG baseline parameters and HRV were finally assessed during visit 3 after the 8-week interventional phase follow-up. During each study-related visit, participants were asked to arrive at the research facility fasted. Due to the COVID-19 pandemic situation, the participants were screened for clinically relevant symptoms before entering the scientific laboratory area. If the participants were thought to be sick or feeling weak, the appointment was rescheduled. 

### 2.3. Fasting Interventions

The participants were enrolled in one of the three fasting interventions.

#### 2.3.1. 16/8. Intermittent Fasting

During the 16/8 fasting intervention, participants (*n* = 11, male 9, female 2) were asked to fast for 16 consecutive hours. During that time, only the consumption of water was allowed. In this context, the consumption of diet drinks, non-sweetened tea or coffee was not allowed. During the other following 8 h, participants were allowed to consume any kind of drink or food of their choice. It was recommended that the participants conducted at least 50% of their fasting period over night-time. 

#### 2.3.2. 20/4. Intermittent Fasting

During the 20/4 fasting intervention, the participants (*n* = 8, male 3, female 5) were asked to fast for 20 consecutive hours. During that time, only the consumption of water was allowed. In comparison to the 16/8 fasting intervention, no diet drinks, non-sweetened tea or coffee were allowed. During the other 4 h, the participants were allowed to consume any kind of drink or food of their choice. Even in this fasting protocol, it was recommended that the participants conducted at least 50% of their fasting period over night-time.

#### 2.3.3. Alternate Day Fasting

During the alternate day fasting period, the participants (*n* = 8, male 4, female 4) were asked to fast for 24 consecutive hours. During that time, only the consumption of water was allowed, with no consumption of alternative drinks, as stated above. On the alternate day, participants were allowed to consume any kind of drink or food of their choice. 

### 2.4. Study Visits

#### 2.4.1. Screening Appointment (Visit 1)

During the screening appointment (visit 1), the course of the trial was introduced to our participating subjects and they were evaluated for their medical anamnesis. The participants’ body compositions were evaluated via bioelectrical impedance assessment (Inbody 720, Inbody Co., Seoul, Republic of Korea) and their body heights were registered manually (Seca 217, Seca, Hamburg, Germany). Additionally, Schellong-1 testing was registered at the laboratory. The testing was performed as follows: initially, the participants had to rest in a supine position for 10 min, followed by standing up as quickly as possible, and finally, a standing period for two additive minutes. The testing was accompanied by Holter-electrocardiograph recording (Holter ECG) (Faros 180; Bittium, Oulu, Finland) to evaluate the sympathicovagal balance and HRV assessment during Schellong-1 testing. Schellong-1 testing is established as a commonly used clinical orthostatic function testing to prove preserved cardiovascular response and to detect orthostatic abnormalities. Baseline ECG parameters (CardioPart 12, Amedtec, Aue-Bad Schlema, Germany) were recorded subsequently during the laboratory visit.

#### 2.4.2. Trial Visits (Visits 2 and 3) 

After the 4-week initial run-in period, the cardiac measurements from the screening visit (visit 1) were repeated during visit 2, including a 12-lead ECG and Holter-ECG during Schellong-1 testing. Participants were instructed during the course of the study, especially the IF periods. Afterwards, the participants’ general health statuses were determined via physical examination. 

Following randomization to the three different IF cohorts, the cardiac measurements were repeated during visit 3. The final visit 3 was conducted in a similar fashion after the 8-week interventional period according to visit 2. Additionally, participants were asked about their general health status via a final physical examination. Participants were supervised by the study team to verify their adherence to each fasting intervention.

### 2.5. ECG Assessment 

The recorded 12-lead ECGs of the participating subjects were analyzed with regard to baseline electrocardiographic parameters, including heart beats at rest (HR) registered in bpm, and ECG time interval measurements, including PQ interval duration (assessed in ms), QRS interval pattern (recorded in ms) and QTc time interval dynamics (measured in ms) [22]. The computerized measurements of baseline ECG parameters were performed digitally using the Amedtec–ECG assessment software (CardioPart 12, Amedtec, Aue-Bad Schlema, Germany) [22]. All study participants were critically evaluated for the following suspected ECG abnormalities: potential clinical relevant sinus bradycardia (predefined as HR < 60 bpm); physiological atrioventricular blocks (AVB), defined as the first and second (Mobitz I) degree AVBs; abnormal right and left axis aberration (defined as more positive than 110 or more negative than 0); pathological QRS interval prolongation (estimated as QRS lengthening > 120 ms) or criteria of preexcitation syndromes, as well as early repolarization (ER) patterns [22]. In the end, no participant had to be eliminated due to abnormal baseline ECG parameters.

### 2.6. HRV Assessment 

The long-term Holter-ECG recording—administrated in our trial—utilized one channel with a 250-Hz sampling frequency [22]. In this context, during the laboratory Schellong-1 testing, the following ECG data measurements were evaluated to assess the sympathicovagal balance based on HRV analyses: firstly, the standard deviation of R-R intervals (SDNN); secondly, the square root of the mean standard difference of successive R-R intervals (RMSSD); and finally, logarithmic analysis referring to the ratio low frequency/high frequency, ln (LF/HF) [22]. As reported in our previous research, a power spectral analysis regarding the frequency domain assessment was performed using Fast-Fourier Transformation in Cardiscope (developed by Hasiba Medical GmbH, Graz, Austria) [22]. The balance of the autonomic nervous system was displayed by the HRV data evaluation of RMSSD and the ratio of low frequency/high frequency (ln (LF/HF)) [22]. Our HRV assessment and data acquisition were conducted based on the following current Task Force guidelines: firstly, the European Society of Cardiology (ESC) guidelines, and secondly, the recommendations of the North American Society of Pacing and Electrophysiology (NASPE) [23,24]. 

### 2.7. Statistics 

All acquired data were processed in SPSS software (IBM SPSS Statistics 28, IBM, New York, NY, USA). Our data were assessed for normal distribution by analyzing the data via the Shapiro–Wilk normality test. Afterwards, we evaluated our data via the analysis of variance testing (ANOVA) referring repeated measurements for interaction differences across our participating subjects. The interaction differences due to variable factors, such as time, phase and group were taken into consideration by two-way ANOVA testing. Adjusted post hoc tests were performed in order to explore the differences between the different group data means. Statistical significance was accepted at *p* < 0.05, whereby the statistical significance of our results was judged by performing an appropriate F statistic in combination with the corresponding *p*-value. 

## 3. Results

A total of 27 healthy people (male 16, female 11) were included in the study. A total of 5 participants of the initially enrolled 32 subjects terminated the study prematurely due to incompliance or gastrointestinal disorders. Hence, the data assessment was based on 27 data sets; the anthropometric characteristics of the equally distributed participating subjects are displayed in Table 1. 

The baseline ECG parameters are presented in Table 2. Baseline ECG parameters, such as HR (measured in bpm), PQ duration (displayed in ms), QRS duration (assessed in ms), and QTc interval analysis (assessed in ms) were analyzed across our three different fasting cohorts in the run-in period (visit 1 to visit 2) as well as after randomization in the intervention period (visit 2 to visit 3). All participants showed comparable findings due to the analyzed 12-lead ECG baseline measurements without any relevant clinical pathology and no significant severe abnormal ECG findings at rest, such as atrioventricular blockings, QRS widening or complete branch blocking, any hints for preexcitation syndrome patterns or ER abnormalities. 

By focusing on the data analyses of HR measurements (in bpm), significant time × group effects for the HR interval assessment could be revealed across the three participating IF cohorts (*p = 0.040*, results displayed in Table 2 and Figure 2 as interaction Δ heart rate differences). Upon detailed consideration, a pronounced Δ heart rate reduction for the 20/4 fasting participants and a mild Δ heart rate reduction in the ADF cohort could be obtained during the intervention period, whereby no relevant Δ heart rate reduction for the 16/8 cohort could be elucidated. These individually pronounced differences between the three different fasting cohorts are displayed in Figure 2.

Additionally, in the data assessment of the QRS interval duration, significant time × group effects could be proven across the three participating IF cohorts (*p* = 0.032, results displayed in Table 2). By detailed statistical consideration, no relevant differences within the intervention phase could be proven in the 16/8 IF cohort, whereby slight—clinically safe—QRS interval widening could be elucidated in the 20/4 IF cohort as well as in the ADF cohort. In this context, it must be stated that the highest—clinically safe—starting level of QRS interval duration was generally observed in the 16/8 IF cohort at the beginning of the intervention phase.

The statistical characteristics displayed in Table 2 provide the progression from visit 1 to visit 2 (run-in period) and from visit 2 to visit 3 (interventional period).

None of the initial analyzed baseline ECG parameters showed significant inter-cohort differences (HR: *p* = 0.336; PQ: *p* = 0.353; QRS: *p* = 0.168; QTc: *p* = 0.160).

The influence of different short-term IF and TRF schedules on the HRV displaying ACM are presented in Table 3. Our data display comparable preserved ACM of the participating subjects in the intervention phase independently of the performed fasting schedule. In this context, significant changes could be elucidated for SDNN (ms) in the 20/4 IF cohort as well as in the ADF cohort in the interventional period (*p* = 0.031, results displayed in Table 3). 

The statistical characteristics displayed in Table 3 provide the progression from visit 1 to visit 2 (run-in period) and from visit 2 to visit 3 (interventional period). 

None of the initially analyzed baseline ECG parameters showed significant inter-cohort differences (SDNN: *p* = 0.516; RMSSD: *p* = 0.871; ln(LF/HF), *p* = 0.752).

The participants’ HRV data were collected over three periods by Holter ECG recording: 10 min in a sitting position, followed by the fast standing up and the final standing duration, as described in our previous research [22].

By discriminating the detailed ECG and ACM variations—in reference to the different time periods during the Schellong-1 testing across our three different IF and TRF schedules—we recorded the differences in participants’ cardiac data in response to the standing-up compared to the sitting period in the controlled run-in period and observational period follow-up. These differences are displayed as Δ measurements in Table 4 and Table 5. 

Therefore, no significant interaction differences could be elucidated in the baseline ECG data assessment during the controlled run-in period and interventional period (the results are displayed in Table 4).

The statistical characteristics displayed in Table 4 provide the progression from visit 1 to visit 2 (run-in period) and from visit 2 to visit 3 (interventional period). 

None of the initial analyzed baseline ECG parameters showed significant inter-cohort differences (ΔHR: *p* = 0.456; ΔPQ: *p* = 0.832; ΔQRS: *p* = 0.302), except ΔQTc: *p* = 0.038.

According to the presented ACM of HRV assessment in Table 3, no significant differences between the different time periods during Schellong-1 testing could be revealed for the presented Δ values of the HRV assessment, thus displaying similar preserved ACM independently of the participants’ fasting schedule (results represented in Table 5).

The statistical characteristics displayed in Table 5 provide the progression from visit 1 to visit 2 (run-in period) and from visit 2 to visit 3 (interventional period). 

None of the initial analyzed baseline ECG parameters showed significant inter-cohort differences (ΔSDNN: *p* = 0.943; ΔRMSSD: *p* = 0.842; Δ ln(LF/HF), *p* = 0.386).

## 4. Discussion

This secondary outcome analysis of a randomized trial with an initial run-in period revealed that different intermitted short-term fasting interventions and TRF protocols in healthy individuals are associated with no significant ECG changes of concern. 

Alterations in baseline ECG parameters, i.e., QT interval alterations and HRV values, are known, relevant, clinical indicators, and small alterations in these parameters might be associated with life-threatening arrhythmogenic events and sudden cardiac death [22,25,26]. Acute increases in blood glucose concentration—within physiological range—have especially been demonstrated to be associated with higher parasympathetic and lower sympathetic cardiac autonomic modulation and subsequent HRV alterations [27]. 

Previous research on the impact of IF and cardiac effects revealed contrary findings. Some studies in healthy individuals demonstrated no increased cardiac risk during body mass reduction by the modified fasting intervention of moderate duration [28] and unaltered ECG following the fastening condition [29]. Contrary findings were revealed previously which analyzed the influence of brief and transient fasting on ECG parameters as well as HRV data, whereas decreased measurements for QRS duration, baseline HR and changes in QTc duration as well as HRV alterations could be revealed [27,30,31]. Additionally, the presence of transient ER pattern, QRS shortening as well as decreased HR have been reported in subjects with food deprivation [32,33]. The substantial impact of fasting on ECG parameters with a certain intersubjective variability of physiological response by food intake, demonstrated in previous research [30], can be confirmed by our data on ECG parameter changes. In this context, our obtained findings in healthy subjects demonstrated the safety of various fasting interventions with no clinically relevant ECG parameter changes. The reduction in baseline HR observed during the run-in period in the 16/8 and ADF group can be explained by the study participants becoming more used to the study environment and potentially regarded as regression to the mean. Our observations did not demonstrate any negative impact of QRS widening nor Δ QTc alterations and adverse cardiac events, such as heart rhythm abnormalities, syncope or proarrhythmogenic risk, but a preserved sympathicovagal balance. Therefore, our findings might provide important information for the feasibility and safety of IF and ACM in healthy young subjects. 

Additionally, the obtained data of our study emphasize maintaining autonomic cardiac modulation due to variable metabolic fasting conditions in our healthy young participants and could not reveal an increased risk for arrhythmogenic substrates and adverse cardiac events determined by the baseline 12-lead ECG assessment. Therefore, the administrated variable IF and TRF schedules in our healthy young participants seem to be safely transferable in daily routines based on our short-term follow-up assessment. Nevertheless, our observations have to be handled with care due to the following reasons: firstly, cardiac repolarization changes in general are known to be individually associated with fasting and feeding; secondly, the obtained QTc levels might be influenced by variable sympathetic activity due to circadian rhythmic; therefore, finally, the ACM might play a role in fasting-related QTc changes [30,34]. In order to obtain comparable and unbiased results, we scheduled the visits (mainly) in the morning hours to exclude any interferential effects of the variable sympathetic activity due to the circadian rhythm and to exclude the previously described QTc shortening after lunch and QTc lengthening after dinner [35].

Our observations, comparing—to the best of our knowledge—primarily ECG and ACM in healthy individuals during three different short-term IF and TRF interventions, could not elucidate clinically relevant QRS widening nor QTc lengthening and sequential proarrhythmogenic risk nor increased risk of sudden cardiac death. Previous case reports displayed a certain risk of QT interval lengthening after fasting or following with very low-calorie diets and its proarrhythmogenic impact by sudden attacks of ventricular torsades de pointes tachycardias [36,37]. Furthermore, changes in metabolic parameters, such as diet habits, are associated in previous case reports with the occurrence of premature ventricular complexes (PVC) and ventricular arrhythmias in subjects without any cardiac and organic diseases. These PVC are usually estimated to be benign in healthy subjects, whereby the homeostasis of myocardial cellular metabolisms and ion channels are essential to maintain electrophysiological stability and to prevent cardiac arrhythmias due to metabolic changes during IF [38].

Globally, cardiovascular disease and associated metabolic disorders have been progressing in recent decades due to the exposure of individuals to suboptimal metabolic and hormonal factors as well as increasing stress levels [18]. In this context, the obtained findings of our pilot trial on different daily practicable IF strategies might contribute to the future prevention of cardiovascular diseases and promote the cardiometabolic health of individuals. Next to regular physical activity, stress-level reducing strategies and conscious daily micro- and macronutrient intake, IF and TRF strategies seem to have great future importance for combating metabolic diseases and the global problem of obesity-related illness [19]. In this context, our pilot study provides important information for safe application in healthy individuals without any enhanced risk of cardiac arrhythmogenic disorders or cardiac stress levels.

However, our preliminary reporting has diverse limitations. First of all, the number of included subjects is relatively small (*n* = 27), for which reason, the obtained data of our pilot study should be considered as hypothesis-generating and further research should focus on a larger sample size of participating subjects to verify the scientific evidence base. By assessing our pilot study results, the following circumstances and their potential impact on ACM and ECG parameters throughout the study period have to be taken into consideration: individual variable differences with regard to physical activity, which have not been determined initially for the participating subjects, as well as lifestyle habits in general during the intervention, such as interindividual sleeping habits, as reported in our previous research [22].

## 5. Conclusions

This secondary outcome analysis of the EDIF trial provided, for the first time, new evidence of the persevered sympathicovagal balance—determined by baseline ECG parameters and HRV assessment—across the three different participating short-term IF and TRF cohorts without any negative clinical impact on the enhanced severe clinical proarrhythmogenic events.

In conclusion, IF and TRF seem to be efficiently transferable into daily routines in healthy young individuals displaying preserved sympathicovagal balance and did not reveal any enhanced cardiac arrhythmogenic potential and cardiac stress levels during our intervention phase. 

## Figures and Tables

**Figure 1 biology-12-00372-f001:**
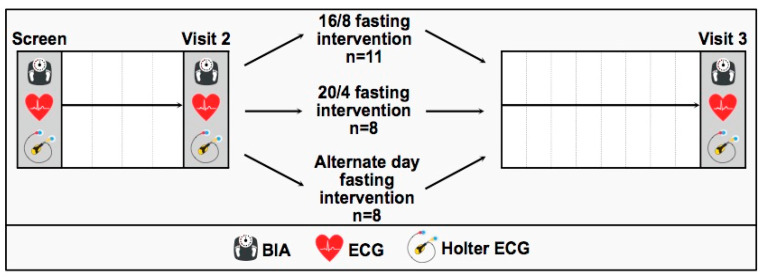
Study flow chart including a controlled run-in period of 4 weeks (screen to visit 2) and an interventional phase follow-up of 8 weeks (visit 2 to visit 3) after randomization to the three fasting interventions: 16/8 IF, 20/4 IF and ADF.

**Figure 2 biology-12-00372-f002:**
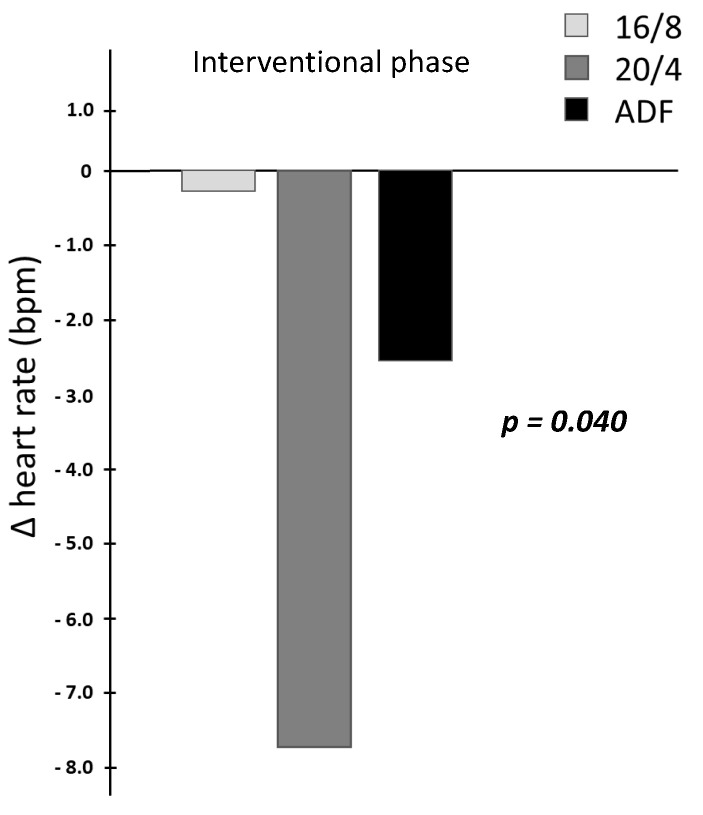
Interaction Δ heart rate differences due to the fasting intervention.

**Table 1 biology-12-00372-t001:** Participants’ Anthropometric Characteristics.

Parameter	Overall (n = 27)	16/8 TRF (n = 11)	20/4 TRF (n = 8)	ADF (n = 8)	*p*-Value
**Age (y)**	26.3 ± 3.8	26.3 ± 4.1	25.8 ± 2.4	25.3 ± 2.1	0.8386
**Body mass (kg)**	77.8 ± 13.9	80.3 ± 18.3	74.4 ± 12.7	78.1 ± 9.1	0.7615
**Height (cm)**	176.9 ± 9.2	177.8 ± 11.1	176.6 ± 7.0	176.4 ± 6.4	0.9386
**BMI (kg/m^2^)**	24.7 ± 3.4	25.2 ± 4.4	23.8 ± 2.8	25.0 ± 2.3	0.7388

Data are presented as a median with standard deviation. By one way ANOVA-testing across the three participating fasting cohorts no significant differences could be obtained. *p* value *****, statistically significant (*p* < 0.05). Abbreviations: y, years; kg, kilogram; cm, centimeter; m^2^, square meter; 16/8 TRF, fasting for 16 h and eating for the remaining 8 h as time-restricted feed periods; 20/4, fasting for 20 h and eating for the remaining 4 h as time-restricted feed periods; ADF, alternate day fasting.

**Table 2 biology-12-00372-t002:** Baseline ECG parameters during run-in and intervention phase due to different fasting intervention.

Parameter	Cohort	Visit 1	Visit 2	Visit 3	*F*-Statistics	*p*-Value
**HR (bpm)**	**16/8**	60.2 ± 8.5	56.1 ± 7.7	55.8 ± 7.1	time: *F*(1,24) = 0.66	0.426
	**20/4**	58.4 ± 4.3	58.9 ± 9.2	50.7 ± 7.5	group: *F*(2,24) = 0.60	0.556
	**ADF**	66.4 ± 12.0	63.5 ± 12.5	60.8 ± 10.2	time × group: *F*(2,24) = 3.68	0.040 *****
**PQ interval (ms)**	**16/8**	194.8 ± 36.6	198.0 ± 44.4	190.5 ± 39.7	time: *F*(1,24) = 3.30	0.082
	**20/4**	176.1 ± 27.0	178.6 ± 25.5	172.8 ±22.1	group: *F*(2,24) = 0.34	0.719
	**ADF**	176.6 ± 29.2	177.6 ± 26.9	178.4 ± 42.0	time × group: *F*(2,24) = 0.84	0.444
**QRS interval (ms)**	**16/8**	96.9 ± 9.2	100.9 ± 12.1	100.9 ± 12.3	time: *F*(1,24) = 0.383	0.542
	**20/4**	91.5 ± 6.9	91.3 ±6.3	92.2 ±6.4	group: *F*(2,24) = 1.00	0.383
	**ADF**	91.0 ±4.3	87.9 ±5.6	90.3 ± 4.1	time × group: *F*(2,24) = 3.97	0.032 *****
**QTc interval (ms)**	**16/8**	376.8 ± 15.7	373.6 ± 25.8	373.3 ± 25.4	time: *F*(1,24) = 0.05	0.821
	**20/4**	389.3 ± 20.9	388.1 ± 28.0	384.1 ± 28.0	group: *F*(2,24) = 0.26	0.772
	**ADF**	390.7 ± 14.2	389.9 ± 12.8	391.2 ± 16.0	time × group: *F*(2,24) = 0.27	0.766

Abbreviations: ECG, electrocardiogram; HR, heart rate; bpm, beats per minute; ms, milliseconds; ADF, alternate day fastening. Data are presented as mean with standard deviation. *p* value *****, statistically significant (*p* < 0.05).

**Table 3 biology-12-00372-t003:** Autonomic Cardiac Modulation (ACM) by Schellong-1 testing.

Parameter	Cohort	Visit 1	Visit 2	Visit 3	*F*-Statistics	*p*-Value
**SDNN (ms)**	**16/8**	99.2 ± 61.1	71.9 ± 27.9	71.3 ± 32.5	time: *F*(1,24) = 5.27	0.031 *****
	**20/4**	73.6 ± 38.5	60.3 ± 29.3	74.9 ± 38.1	group: *F*(2,24) = 1.03	0.371
	**ADF**	82.8 ± 35.7	72.9 ± 44.2	82.9 ± 45.4	time × group: *F*(2,24) = 0.05	0.953
**RMSSD (ms)**	**16/8**	84.0 ± 52.8	75.3 ± 35.4	74.9 ± 24.9	time: *F*(1,24) = 0.44	0.513
	**20/4**	72.7 ± 38.8	72.8 ± 48.1	78.9 ±41.0	group: *F*(2,24) = 0.51	0.607
	**ADF**	77.8 ± 44.3	78.8 ±53.8	88.4 ±45.4	time × group: *F*(2,24) = 0.01	0.995
**Ln(LF/HF) (-)**	**16/8**	0.41 ± 1.58	0.10 ± 1.20	−0.17 ± 1.51	time: *F*(1,24) = 3.02	0.095
	**20/4**	0.22 ± 0.82	−0.30 ± 0.81	−0.15 ±0.79	group: *F*(2,24) = 0.14	0.875
	**ADF**	0.73 ± 1.41	−0.04 ± 0.87	0.09 ± 1.50	time × group: *F*(2,24) = 0.74	0.490

Abbreviations: SDNN, standard deviation of normal-to-normal beat; RMSSD, root mean square of successive differences; LF, low frequency; HF, high frequency; ms, milliseconds; ADF, alternate day fastening. Data are presented as mean with standard deviation. *p* value *, statistically significant (*p* < 0.05).

**Table 4 biology-12-00372-t004:** Differences of baseline ECG Parameters due to Standing up Period during Schellong-1 testing.

Parameter	Cohort	Visit 1	Visit 2	Visit 3	*F*-Statistics	*p*-Value
**ΔHR (bpm)**	**16/8**	16.4 ± 12.0	16.4 ± 6.0	15.8 ± 6.7	time: *F*(1,24) = 0.34	0.567
	**20/4**	16.5 ± 4.8	14.8 ± 9.2	20.1 ± 8.6	group: *F*(2,24) = 1.77	0.193
	**ADF**	11.8 ± 5.7	15.7 ± 8.5	17.5 ± 3.8	time × group: *F*(2,24) = 1.19	0.321
**ΔPQ interval (ms)**	**16/8**	−7.0 ± 14.8	−13.4 ± 23.5	−8.6 ± 19.2	time: *F*(1,24) = 1.27	0.272
	**20/4**	−6.5 ± 12.6	−8.4 ± 6.9	−5.2 ± 11.6	group: *F*(2,24) = 0.12	0.885
	**ADF**	−9.9 ± 7.5	−8.8 ± 10.2	−10.9 ± 15.6	time × group: *F*(2,24) = 1.20	0.320
**ΔQRS interval (ms)**	**16/8**	1.1 ± 10.8	−2.4 ± 2.2	−2.9 ± 1.8	time: *F*(1,24) = 0.04	0.838
	**20/4**	3.0 ± 9.7	−0.1 ± 3.5	−1.0 ± 6.1	group: *F*(2,24) = 0.20	0.819
	**ADF**	−3.7 ± 0.8	−2.9 ± 1.2	−5.5 ± 4.6	time × group: *F*(2,24) = 0.73	0.494
**ΔQTc interval (ms)**	**16/8**	0.1 ± 9.0	7.4 ± 8.3	9.1 ± 7.0	time: *F*(1,24) = 0.62	0.440
	**20/4**	2.4 ± 7.8	3.3 ± 11.7	14.9 ± 14.7	group: *F*(2,24) = 0.35	0.706
	**ADF**	−10.6 ± 13.5	−0.5 ± 13.7	−5.9 ± 29.9	time × group: *F*(2,24) = 2.85	0.077

Abbreviations: ECG, electrocardiogram; HR, heart rate; bpm, beats per minute; ms, milliseconds; Δ, delta; ADF, alternate day fastening. Data are presented as mean with standard deviation. *p* value *****, statistically significant (*p* < 0.05).

**Table 5 biology-12-00372-t005:** Differences of HRV Parameters due to Standing up Period during Schellong-1 testing.

Parameter	Cohort	Visit 1	Visit 2	Visit 3	*F*-Statistics	*p*-Value
**ΔSDNN (ms)**	**16/8**	−0.5 ± 44.7	36.4 ± 59.1	31.7 ± 48.5	time: *F*(1,24) = 3.78	0.064
	**20/4**	0.4 ± 45.8	34.3 ± 41.9	39.6 ± 59.7	group: *F*(2,24) = 0.24	0.788
	**ADF**	−6.2 ± 35.4	16.8 ± 49.9	12.0 ± 48.9	time × group: *F*(2,24) = 0.08	0.925
**ΔRMSSD (ms)**	**16/8**	−32.4 ± 38.3	−29.4 ± 38.6	−27.4 ± 27.0	time: *F*(1,24) = 0.55	0.464
	**20/4**	−42.0 ± 32.6	−33.3 ± 38.3	−39.4 ± 34.6	group: *F*(2,24) = 0.68	0.515
	**ADF**	−34.7 ± 35.9	−38.1 ± 48.1	−50.7 ± 41.3	time × group: *F*(2,24) = 0.14	0.873
**ΔLn(LF/HF) (-)**	**16/8**	0.91 ± 1.21	1.68 ± 1.06	1.94 ± 1.25	time: *F*(1,24) = 2.77	0.109
	**20/4**	1.40 ± 1.20	1.99 ± 1.30	2.09 ± 1.00	group: *F*(2,24) = 0.78	0.468
	**ADF**	0.63 ± 0.81	1.83 ± 1.00	2.06 ± 1.03	time × group: *F*(2,24) = 0.15	0.866

Abbreviations: SDNN, standard deviation of normal-to-normal beat; RMSSD, root mean square of successive differences; LF, low frequency; HF, high frequency; ms, milliseconds; Δ, delta; ADF, alternate day fastening. Data are presented as mean with standard deviation. *p* value *****, statistically significant (*p* < 0.05).

## Data Availability

The data are available upon reasonable request.

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
