# Peer review of "Effects of Different Fasting Interventions on Cardiac Autonomic Modulation in Healthy Individuals: A Secondary Outcome Analysis of the EDIF Trial"

_biology, 2023, doi:10.3390/biology12030372_

Round 1

Reviewer 1 Report

Journal: Biology (ISSN 2079-7737)

Manuscript ID: biology-2223243

General Comments

Thank you for giving me the opportunity to review the article: “Effects of Different Fasting Interventions on Electrocardiography and Heart Rate Variability in Healthy Individuals: A Secondary Outcome Analysis of the EDIF trial.” With the increasing international rates on obesity, the subject of this article is of great importance since it considers the relevant and topical issue of Fasting Interventions. The study explores the effect of different fasting intervention on ECG and HRV analyses on 27 healthy adults. The reviewer commends the researchers for the vast amount of work and the that has gone into this research.

The paper seems to be more directed to a clinical (and more specifically cardiovascular) audience and in some instances reads more complicated than is required. However, the reviewer’s suggestion is that the article be accepted after minor revisions.

Additional information is outlined below:

Specific Minor Comments

Abstract 

It would benefit the abstract to add a line about the implications/impact of this study.

Introduction

Reference is made to the EDIF study. It is suggested that the study is written in full before abbreviation, and at least a summary sentence (or two) are provided as a background for any readers not familiar with this work.

It is suggested that the author provides some brief information about the relevance of ECG and HRV parameters in regards to health, or the significance of investigation these factors in the current study.

General

Line 86 and 96: Please check if 0/100% is the same for 8/16 and 20/4 IF.

Figure 1: It is suggested that the legibility of this figure is improved.

A section on Impact and implications of the research would benefit the paper. More information than what is currently provided is required.

Limitations of this study

The authors rightfully stated that a small number of participants (n=27) is a limitation of this study. But with even smaller group numbers (n=8 and n=11) maybe this should be titled as a pilot study?

THE END

Reviewer 2 Report

This work used different fasting protocols 27 showed comparable cardiac autonomic response determined by electrocardiogram and heart rate 28 variability measurements. IF and TRF seem to be safely applicable during daily routine in healthy 368 young subjects with preserved sympathicovagal balance and without enhanced cardiac 369 arrhythmogenic potential and cardiac stress level during our intervention phase. My suggestion is to increase the number of participants. for example, During 20/4 fasting intervention, only 8  participants (n=8, male 3, female 5) were asked to fast for 150 20 consecutive hours. 

Reviewer 3 Report

This manuscript was a secondary analysis of cardiovascular parameters in subjects exposed to different fasting interventions. The findings are straightforward. There a few items that need clarity.

1. There is no relevance to fasting for weight loss. Subjects in all groups had a normal BMI. This is misleading in the introduction, lines 65-82 needs to clarify weight loss studies from other fasting studies.

2. Post hoc tests were not reported for ECG data (data for table 2 or Figure 2). There are significant time X group effects for HR and QRS interval that is not further analyzed or clarified. From Figure 2, there is an intervention effect, but is that just for ADF? 

3. Figure 1 is too small. Text should be removed from the figure and described in the figure caption.

4. Line 108-110 is a run-on sentence that is not clear.

5. What were the objective or trial of EDIF trial? What does EDIF stand for?         

Round 2

Reviewer 2 Report

The manuscript has been revised well.